# A Novel Machine Learning Algorithm to Automatically Predict Visual Outcomes in Intravitreal Ranibizumab-Treated Patients with Diabetic Macular Edema

**DOI:** 10.3390/jcm7120475

**Published:** 2018-11-24

**Authors:** Shao-Chun Chen, Hung-Wen Chiu, Chun-Chen Chen, Lin-Chung Woung, Chung-Ming Lo

**Affiliations:** Department of Ophthalmology, Taipei City Hospital, Taipei 10632, Taiwan; hwchiu@tmu.edu.tw (H.-W.C.); dac03@tpech.gov.tw (C.-C.C.); dac91@tpech.gov.tw (L.-C.W.); buddylo@tmu.edu.tw (C.-M.L.)

**Keywords:** artificial neural network, diabetic macular edema, machine learning, ranibizumab

## Abstract

Purpose: Artificial neural networks (ANNs) are one type of artificial intelligence. Here, we use an ANN-based machine learning algorithm to automatically predict visual outcomes after ranibizumab treatment in diabetic macular edema. Methods: Patient data were used to optimize ANNs for regression calculation. The target was established as the final visual acuity at 52, 78, or 104 weeks. The input baseline variables were sex, age, diabetes type or condition, systemic diseases, eye status and treatment time tables. Three groups were randomly devised to build, test and demonstrate the accuracy of the algorithms. Results: At 52, 78 and 104 weeks, 512, 483 and 464 eyes were included, respectively. For the training group, testing group and validation group, the respective correlation coefficients were 0.75, 0.77 and 0.70 (52 weeks); 0.79, 0.80 and 0.55 (78 weeks); and 0.83, 0.47 and 0.81 (104 weeks), while the mean standard errors of final visual acuity were 6.50, 6.11 and 6.40 (52 weeks); 5.91, 5.83 and 7.59; (78 weeks); and 5.39, 8.70 and 6.81 (104 weeks). Conclusions: Machine learning had good correlation coefficients for predicating prognosis with ranibizumab with just baseline characteristics. These models could be the useful clinical tools for prediction of success of the treatments.

## 1. Introduction

Diabetic macular edema (DME) is a major complication of diabetic retinopathy. The prevalence rate is 2–30% and increased risk in the population with poor diabetes control or longer diabetes years. It was characterized by thickening in the center of macula. The macula is the major area for vision quality. DME causes a severe vision disturbance and is a major cause of the blindness associated with the increasing prevalence of diabetes around the world [1,2]. In addition, diabetes-related and ocular comorbidities are more prevalent in patients with DME than in those without [3]. DME is associated with an ischemic change of retinal blood vessels that results in their release of a large volume of vascular endothelial growth factor (VEGF) [4,5]. These agents make the blood-retinal barrier disrupted and lead to serum or fluid accumulated. Optical coherence tomography (OCT) is a noninvasive and noncontact machine and provides qualitive and quantities information. The advantage of OCT makes it as a multifunctional tool to diagnosis and follow the progression of disease or the responses after treatments. According to the pathogenesis of DME, anti-VEGF agents, including aflibercept, bevacizumab and ranibizumab, are the most effective and safe choices for DME treatment. These medications can capture the VEGF agents, stop the serum or fluid leakage, reverse the thickening of macula and improve vision. More frequent injection of anti-VEGF agents is associated with better control over DME and improved visual acuity [6,7,8,9]. After one year of treatment, the mean vision improvement are 13.3 letters with aflibercept, 11.2 letters with ranibizumab and 9.7 letters bevacizumab. In Taiwan, due to the national insurance only allowed for five or eight injection for whole life, these agents are usually used as rescues and not significantly increasing vision outcomes. However, the risk of complications increases with the number of injections. Due to the complicated pathogenesis and multi-factors related prognosis, until now, there is still no a reliable prognostic factor to predict the final treatment outcome before any injection [10,11]. In this case, if we can establish a method to predict final visual outcome and customized the treatment plan, we can achieve a great improvement in the quality of care in the patients of DME. 

Machine learning algorithms are increasingly being used for ophthalmology applications. Machine learning can use various approaches to analyze and summarize complex datasets for discovery of new knowledge [12,13]. Linear modelling, such as multiple regression analysis, often performs poorly when relationship between variables are nonlinear, as often observed in clinical settings [14]. Similar to the functioning of the human brain, artificial neural networks (ANNs) are comprised of different layers of “neurons” that are interconnected based on numerical weights. Each layer is trained using a regression analysis. Periodically, non-essential items are detected for removal using a validation set. In this manner, learning, thinking and testing are performed to optimize the weights to yield the best network [15]. ANNs can effectively manage a massive amount of information and use nonlinear modeling for calculation. The develop of ANNs by the collection of randomly chosen hidden units and analytically defined output weights make it can precisely predict the unknown relationship between input and output factors through repeated learning, validation and testing is continued until an acceptable regression is obtained [16,17,18]. Using ANNs but not linear statistical methods, Mehra predicted the pre-planting risk of stagonospora nodorum blotch, the optimal growth and culture conditions for maximum biomass accumulation and the incidence of dengue fever [19]. In addition, the use of ANNs can distinguish photographs of diabetic retinopathy and macular edema with retinal fundus [20,21,22]. Moreover, the accuracy rate could achieve up to 94.5% by deep convolutional neural network [23].

The National Institutes of Health-sponsored Diabetic Retinopathy Clinical Research Network (DRCR.net) provides a database of multicenter clinical research on diabetic retinopathy. Protocol I in the DRCR.net database involves a study on the use of an intravitreal injection of ranibizumab or triamcinolone acetonide in combination with laser photocoagulation for diabetic macular edema (DME) [7,24,25]. Here, in patients with DME that were treated with an intravitreal injection of ranibizumab, we use baseline patient characteristics to develop and test a machine learning algorithm to predict long-term visual outcomes. We used ANNs to build decision-support models to predict visual acuity in patients with DME at 52, 78 and 104 weeks after ranibizumab treatment.

## 2. Patients and Methods

For algorithm development and taking advantage of publicly available patient data through the National Institutes of Health-sponsored DRCR.net, which provides a database of multicenter clinical research on diabetic retinopathy treated in the USA, we retrospectively analyzed data from patients treated with the Protocol I in the DRCR.net database [7,24,25]. Protocol I, is a multicenter clinical trial that evaluated the use of an intravitreal injection of ranibizumab or triamcinolone acetonide in combination with laser photocoagulation for DME.

The dataset included information from participants who were followed up for 5 years to evaluate the long-term effects of ranibizumab. Based on the one-year results of the study, in April 2010, all participants regardless of randomization group were eligible to receive ranibizumab treatment. The initial 127 datasets included 674 patients who were administered an intravitreal injection of ranibizumab, corticosteroids, or vehicle. Only 454 patients received ranibizumab and were followed up for more than 52 weeks. Since this data was obtained from a publicly available data base (DRCR.net), our study was exempt from the requirement of informed patient consent.

### 2.1. Prediction Model Input and Target Output

Due to missing data, we used the patient data at 52, 78 and 104 weeks, which means one, one and half and two years. These three time points are the most used to comparing the effectiveness of DME treatments and following the progression of DME. The longest following duration in the dataset were two years. The input variables to build a prediction model were sex, age, diabetes type, insulin use, glycated hemoglobin (HbA1c) level, hypertension under treatment, hypercholesterolemia under treatment, lens status, degree of diabetic severity, the baseline macular OCT value (central point, central, inner and outer superior/nasal/inferior/temporal part), the timetable of ranibizumab treatment and baseline visual acuity (Table 1). Visual acuity at 52, 78 and 104 weeks was used to define the target output. The target outputs were continuous as early treatment diabetic retinopathy study (ETDRS) letters. 

### 2.2. Machine Learning Development

To build the ANN, a complex learning procedure is used to identify data patterns. Once the algorithm is established, the ANN will verify the correlation coefficients of the algorithm through testing. Training group data were used for establishing and training the ANN models and validation and testing group data were used to evaluate the predictive performance of the well-trained models. In this manner, the results of training, validation and testing can sufficiently demonstrate the good generalization ability of the trained neural network. After these steps, the ANNs select the best network structure or algorithm with the most satisfactory performance. After this strict process of development, the algorithm can be applied to support a medical decision. We used a multiple-layer perceptron (MLP) model with a back propagation learning rule. This network model was composed of one input layer, one hidden layer and one output layer (Figure 1). The input layer was composed of the input variables and the output layer is the final visual outcome. Computation is performed on the hidden layer to establish the best connection between the input and output layers. The ANN models were developed using the ANN tool embedded in Statistica 10.0 (StatSoft, Inc., Tulsa, OK, USA). This tool also provided the embedded trial-and-error procedures with a changing number of hidden layers to produce the potential models for use. Cross validation was applied using 80% of the training group, 10% of the validation group and 10% of the testing group to determine the generalization of the models. Data subsets were chosen by randomly sampling from the set of all information. Correlation coefficients were determined for each MLP model. The mean standard error (SEM) reported was the standard error of the final visual outcome and the SEM was in reference to the ETDRS letter. The sensitivity analysis links the weight of each input variable to each MLP model. The greater the number, the greater the effect of that input feature. This ANN design did not allow for statistical significance values.

## 3. Results

The most influential baseline parameters on post-treatment visual acuity were baseline visual acuity, lens status and the intravitreal injection (IVI) time table at 52, 78 and 104 weeks (Table 2). 

For the target of visual acuity at 52 weeks, the MLP model had 58 input features, 21 hidden neurons and 1 output features (i.e., MLP 58-21-1). The correlation coefficients and SEM for the training, testing and validation groups were 0.75, 0.77 and 0.70 and 6.50, 6.11 and 6.40 ETDRS letters, respectively. The most related input variables ranked by sensitivity analysis for all three groups were visual acuity at baseline (2.07), lens status (1.44) and the application of injection at the fourth week (Figure 2A).

For the visual acuity at 78 weeks target, MLP 72-48-1 was observed. The correlation coefficients and SEM for training, testing and validation groups were 0.79, 0.80 and 0.55 and 5.91, 5.83 and 7.59 ETDRS letters, respectively. The most related input variables ranked by sensitivity analysis for all three group were VA at baseline (2.20), lens status (1.77) and IVI time table at 44th week (Figure 2B).

For the target VA at 104 weeks, the ANN name was MLP 84-21-1. The correlation coefficients of the training, testing and validation groups were 0.83, 0.47 and 0.81. The SEMs for the different time point group were 5.39, 8.7 and 6.81 ETDRS letters. The most related input variables ranked by sensitivity analysis for all three group were visual acuity at baseline (2.60), lens status (2.19) and IVI time table at week 12. Figure 2C shows the regression model for visual acuity at 52, 78 and 104 weeks.

## 4. Discussion

Machine learning algorithms have been used for the prognosis of cancer, post-traumatic stress disorder, to detect brain pathology and to predict survival in patients with burns [26,27,28,29]; however, our study is the first to describe a novel machine learning algorithm that predicts visual outcomes in patients with diabetic macular edema who are treated with intravitreal ranibizumab.

We used patient baseline characteristics and ANN machine learning to build an algorithm that successfully predicts long-term visual acuity outcome as ETDRS letters in patients with DME who are treated with intravitreal ranibizumab. The final SEM of visual acuity prediction at 52, 78 and 104 weeks were 6.3, 6.4 and 7.0 ETDRS letters. The high correlation coefficients and low SEM of our results demonstrate that our automated system is a useful tool for informing treatment choice in patients with DME. 

Several studies have investigated approaches for predicting DME treatment outcomes after IVI anti-VEGF therapy and report that an important factor is visual acuity at baseline [7,20,30,31,32,33]. Our study using machine learning also found that baseline visual acuity had a significant role in predicting outcomes at three different time points. These results support the idea that early detection and aggressive treatment can improve visual outcomes in patients with DME. We also found that lens status is an important consideration in the prediction of visual treatment outcome. This finding may be related to the IVI procedure increasing the progression of cataract [34]. The progression of cataract in phakic eyes after IVI may influence the final visual outcome. 

The IVI number is critically important for the treatment of DME. Many studies suggest that more frequent IVI injections yield better final visual acuity outcomes [6,7,8,9]. However, IVI is sometimes cost prohibitive and not suitable for all clinical settings [35]. In the present study, we use the IVI time points as pre-treatment input instead of the total numbers of injections, to ensure the general applicability of our algorithm. 

In contrast to previous reports, we did not find that baseline OCT parameters influenced visual outcomes. This difference is likely because we only included baseline OCT measurements in our models, while prior studies also included follow-up OCT measurements [30,32,36,37].

Our study has several limitations. Protocol I used a focal laser adjunct treatment for DME. However, we did not include this variable as input because the power setting and treatment protocol for the focal laser treatment was too variable. Other limitations were the small number of patients in our study and the limited combination of treatment protocols. As a result, our machine learning approach may not predict the best IVI protocol for every patient. In the future, we will combine the protocol I data with data from other clinical settings to build a more elegant algorithm. In addition, we will use patient baseline variables and machine learning to predict the most efficient time table of treatment. The ratio of insulin using in the protocol I was higher than the real world. And it may not reflect the outcomes of patients using oral medications. It must be noted that adverse effects associated with bevacizumab administration in the protocol I were very rare, not more than 10% (include the corresponding reference here), therefore we did not include the variable adverse effects in our algorithm.

## 5. Conclusions

Using only patient baseline clinical characteristics data, we can create algorithms based on ANNs with good correlation coefficients for predicting final visual acuity at 52, 78 and 104 weeks in intravitreal ranibizumab-treated patients with DME. The SEM was 5–9 ETDRS letters or approximately 1–2 lines of vision. Our models may be useful in the clinic as tools aiding expectation and explanation. Further research using machine learning may improve the care and outcomes for patients with DME.

## Figures and Tables

**Figure 1 jcm-07-00475-f001:**
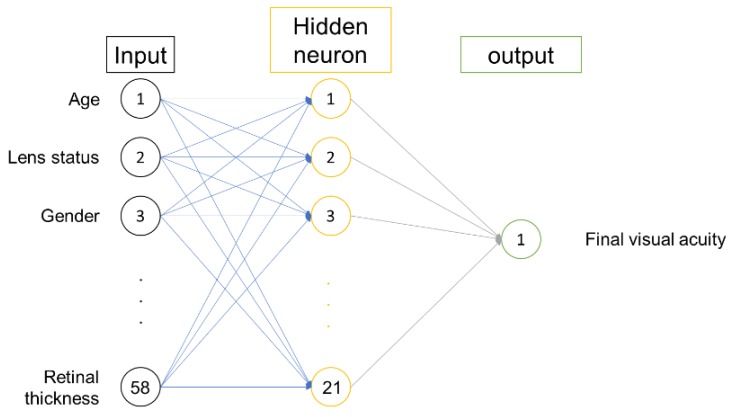
The artificial neural network model. This network model was composed of one input layer, one hidden layer and one output layer. The input layer was composed of the input variables and the output layer is the final visual outcome in letters.

**Figure 2 jcm-07-00475-f002:**
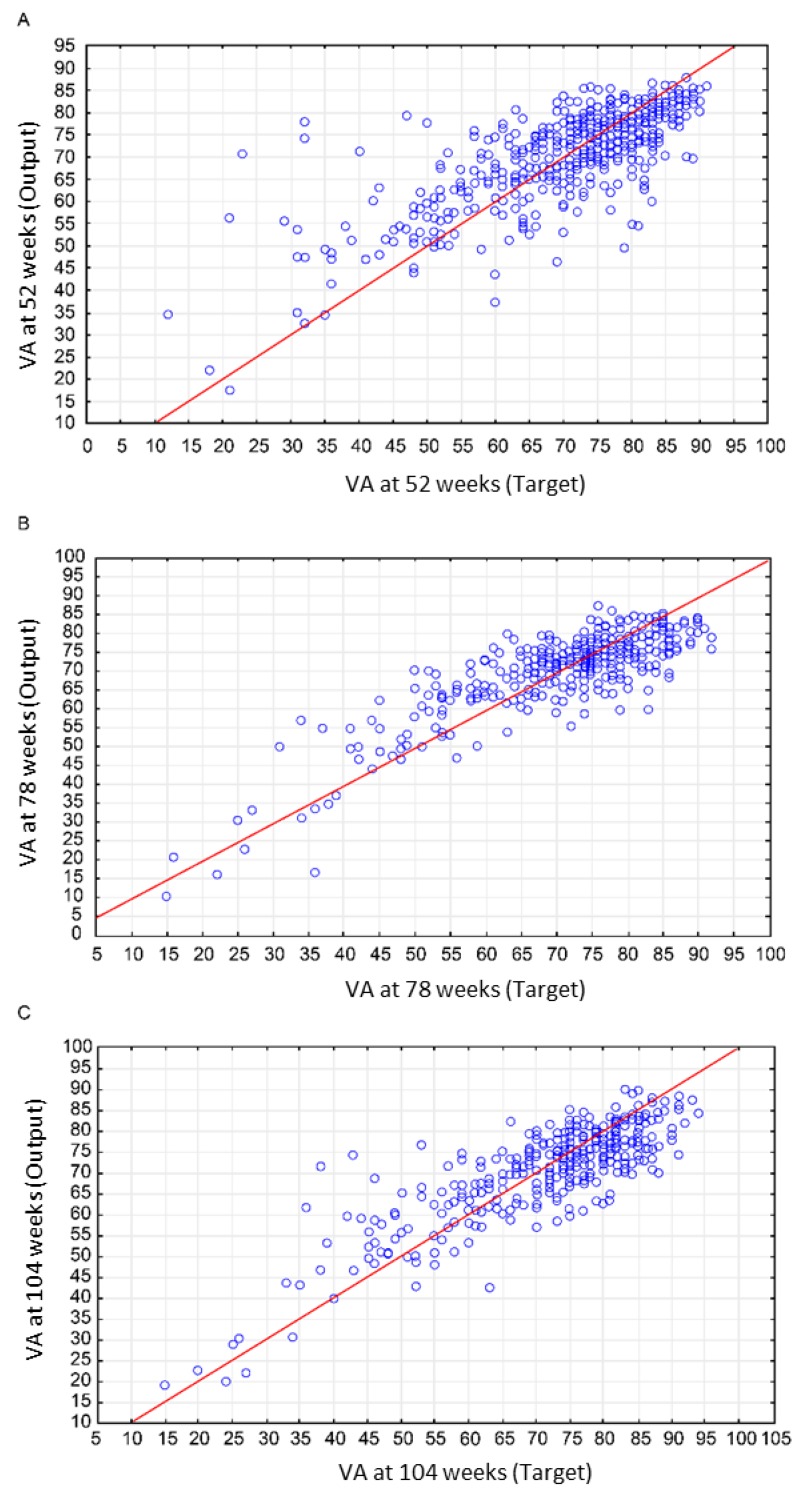
Final visual acuity prediction using machine learning with baseline inputs variables. Scatter plot of output vs. target (ETDRS letters): **A**. 52 weeks, **B**. 78 weeks and **C**. 104 weeks. Red line: perfect correlation regression line. The red lines represent the perfect lines without error predication. VA, visual acuity.

**Table 1 jcm-07-00475-t001:** Patient characteristics at different time points after treatment.

Characteristics	52 Weeks	78 Weeks	104 Weeks
No. of study eyes	512	483	464
Sex, male No. (%)	290 (56.6)	272 (53.3)	262 (56.5)
Age, mean (SD) (years)	62.6 (9.5)	62.5 (9.5)	62.6 (9.5)
Diabetes			
Type 2 No. (%)	468 (91.4)	442 (91.5)	425 (91.6)
HbA1c level, (SD)	7.67 (1.55)	7.66 (1.52)	7.66 (1.55)
Insulin using No. (%)	308 (60.2)	291 (60.2)	278 (59.9)
Comorbidities under treatment, No. (%)			
Hypertension	404 (78.9)	383 (79.3)	367 (79.1)
Hyper cholesterol	350 (68.4)	333 (68.9)	320 (69.0)
Lens status, pseudophakic No. (%)	154 (30.1)	149 (30.8)	139 (30.0)
Diabetic retinopathy severity, No. (%)			
Microaneurysms only	16 (3.1)	15 (3.1)	15 (3.2)
Mild/moderate NPDR	266 (52.0)	250 (51.8)	241 (51.9)
Severe NPDR	105 (20.5)	96 (19.9)	93 (20.0)
PDR and/or prior scatter	123 (24.0)	120 (24.8)	113 (24.4)
Visual acuity with ETDRS letter, mean (SD)			
Baseline	63.6 (12.5)	63.2 (12.4)	63.2 (12.4)
Final	70.4 (13.5)	70.2 (13.5)	70.6 (13.9)
Intra-vitreous injection No. (SD)	8.1 (2.7)	10.2 (4.2)	11.7 (5.5)
Retina thickness of grid (um) (SD)			
Center Point	391.6 (135.9)	394.8 (136.5)	396.7 (138.1)
Center subfield	392.7 (122.6)	395.4 (123.6)	397.3 (125.0)
Inner/outer subfield			
Superior	358.2 (92.0)/291.6 (72.0)	359.9 (93.4)/292.3 (72.3)	360.8 (94.2)/292.6 (73.1)
Nasal	359.8 (92.2)/299.5 (66.9)	361.2 (93.5)/300.0 (67.0)	362.3 (94.4)/300.4 (67.7)
Inferior	364.4 (102.7)/286.6 (77.1)	365.6 (104.3)/287.3 (78.3)	366.5 (105.2)/287.3 (78.6)
Temporal	368.0 (104.1)/289.1 (82.7)	370.0 (105.9)/290.3 (84.4)	370.7 (106.7)/290.1 (84.6)

SD, standard deviation; HbA1c, glycated hemoglobin; NPDR, nonproliferative diabetic retinopathy; PDR, proliferative diabetic retinopathy; ETDRS, early treatment diabetic retinopathy study.

**Table 2 jcm-07-00475-t002:** Machine learning prediction of final visual acuity.

Weeks	Net Name	Correlation Coefficients			Mean Standard Error (ETDRS Letters)		
		Train Group	Test Group	Validation Group	Train Group	Test Group	Validation Group
52	MLP 58-21-1	0.75	0.77	0.70	6.50	6.11	6.40
78	MLP 72-48-1	0.79	0.80	0.55	5.91	5.83	7.59
104	MLP 84-21-1	0.83	0.47	0.81	5.39	8.70	6.81

MLP, multiple-layer perceptron; ETDRS, early treatment diabetic retinopathy study.

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
