# Peer review of "A Novel Machine Learning Algorithm to Automatically Predict Visual Outcomes in Intravitreal Ranibizumab-Treated Patients with Diabetic Macular Edema"

_jcm, 2018, doi:10.3390/jcm7120475_

Reviewer 1 Report

The authors propose an artificial  neural network approach to predicts prognosis after ranibizumab treatment.The manuscript is well written and organised. The technical proposal is sound and supported by experimental results. However, references should be properly linked with corresponding statements/sentences in the main text that is a real concern, please fix that issue.

Moreover, I do strongly suggest the authors take into account other works on machine learning that can provide a wider scope to the introduction, which in turn might better demonstrate the key role of novel artificial neural network algorithms in everyday applications, such as extreme artificial neural networks, and deep neural networks.

The authors are strongly encouraged refer to the two works below reported in their introductory section:

Salerno, V.M.; Rabbeni, G. An Extreme Learning Machine Approach to Effective Energy Disaggregation. Electronics 20187, 235.

Xu, K.; Feng, D.; Mi, H. Deep Convolutional Neural Network-Based Early Automated Detection of Diabetic Retinopathy Using Fundus Image. Molecules 201722, 2054.

Author Response

Dear Mr/Mrs. Reviewer

Thank you for your hard working for this paper. I had to give all my pressure to your brain burning.

I had add "Valerio" working in the introduction line 73 and replace original reference 17. 

Also, I had add "Xu" working inthe introduction line 81 and replace original reference 20.

I also corrected the wrong position of references.

Thank you for your hard working and hope you can revew it again.

Best regards.  

Reviewer 2 Report

line 177 correction:  who "are" treated with ranibizumab

A well written paper.  

Please add further clarification as to exactly what "IVI time table at the fourth week" means as a variable.  This in confusing to the reader and is of key importance.  Otherwise looks good.

Author Response

Dear Mr/Mrs. Reviewer

Thank you for your hard working for this paper. I had to give all my pressure to your brain burning.

I had corrected the wrong style in the line 177.

Also, I had change the term "IVI time table at the fourth week" to "the application of injection at the fourth week" to more clearify the importance of the rold of injection in that time point.

Thank you for your reviewing again after these corrections.

Best regards.